# A Comparative Study of Unsupervised Adversarial Domain Adaptation Strategies in Multiple-instance Learning Frameworks for Digital Pathology

**Javier Garcia-Baroja**[1]                                JAVIER.G.BAROJA@GMAIL.COM

**Samaneh Abbasi-Sureshjani**[2]                            SAMANEH.ABBASI@ROCHE.COM

**Nazim Shaikh**[2]                                        NAZIM.SHAIKH@ROCHE.COM

**Konstanty Korski**[2]                                    KONSTANTY.KORSKI@ROCHE.COM

[1] *Swiss Federal Institute of Technology,Rämistrasse 101, 8092 Zürich*

[2] *F. Hoffmann-La Roche AG, Grenzacherstrasse 124, 4070 Basel, Switzerland*

**Editors:** Accepted for publication at MIDL 2023

## Abstract

Performance of state-of-the-art deep learning methods is often impacted when evaluated on data coming from unseen acquisition settings, hindering their approval by the regulatory agencies and incorporation to the clinic. In recent years, several techniques have been proposed for improving the generalizability of models by using the target data and their corresponding ground truths. Some of those approaches have been adopted in histopathology, however they either focus on pixel-level predictions or simple tile level classification tasks with or without target labels. In this work, we investigate adversarial strategies in weakly supervised learning frameworks in digital pathology domain without access to the target labels, thereby strengthening the generalizability to unlabeled target domains. We evaluate several strategies on Camelyon dataset for metastatic tumor detection tasks and show that some methods can improve the average F1-score over 10% for the target domain.

## 1. Introduction

Despite the popularity of computer aided diagnosis tools for Digital Pathology (DP), widespread use of these algorithms is hampered by the inherent variation between images of diverse origin (Howard et al., 2021) (in staining, thickness, patient demographics, etc.) known as *domain shift*. Therefore, a strategy that enables us to build more generalizable models is desired. Among different methods, Marini et al. (2022) propose a Domain Adversarial Neural Network (DANN) (Ganin et al., 2015) to tackle stain heterogeneity with an understanding of domain rooted in Whole Slide Image (WSI) coloring. The Conditional Domain Adversarial Network (CDAN) proposed by Long et al. (2017) also facilitates domain alignment by utilizing the discriminative information offered by main-task classifier predictions.

This paper focuses on Unsupervised Domain Adaptation (UDA) and addresses domain shift caused by scanner variations in weakly supervised metastatic tumor detection. We explore DANN, CDAN, and the impact of changing the position of the domain discriminator in attention MIL and TransMIL (Shao et al., 2021) networks.

## 2. Methods and Experimentations

We propose to adapt MIL models by combining the discriminators ($\mathcal{G}$) in DANN and CDAN at two locations: 1) after a shallow encoder ($loc_i$), where $\mathcal{G}$ would receive *instance-level* samples. In this way, the feature alignment between domains will be provided by a shallow encoder that maps the embeddings from the frozen encoder into an overlapping latent space; 2) $\mathcal{G}$ is positioned after the embedding aggregation step ($loc_b$), that is, after the attention mechanism in attention-MIL or mean pooling of patch tokens after the last transformer layer in TransMIL. This ensures domain alignment on the *aggregated instances* that are forwarded to the final slide-level classifier. The adapted MIL pipelines for DANN and CDAN are depicted in Figure $1(a)$ and $1(b)$, illustrating each integration location.

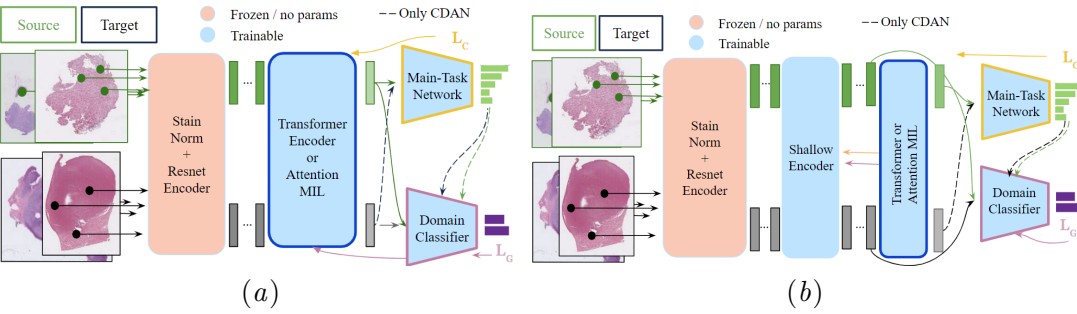

Figure 1: Overview of the two UDA approaches. a) shows $loc_b$ and b) $loc_i$.

### 2.1. Experimental Setup

The experiments used a combination of the publicly available Camelyon16 and Camelyon17 datasets (Litjens et al., 2018), which contains 1399 WSI of lymph nodes (metastatic and healthy) stained with Hematoxylin and Eosin, from three different scanners, five hospitals.

Scanner 1 (S1) data (from three different medical centers digitized by the same scanner) was used as the source ($N_s = 544$), while Scanner 2 (S2) data (from two hospitals) and Scanner 3 (S3) data (from one facility) comprised the target dataset ($N_{t_2} = 253$ and $N_{t_3} = 100$), on which the model is to improve its generalizability. The source dataset was split into 5 non-overlapping subsets (each 20%) stratified by medical center and tumor label, for 5-fold Cross-Validation (CV). The UDA training had S2 as target and was evaluated on S3.

10,000 tiles at $40\times$ magnification were extracted from each WSI. Image patches were stain normalized by Tellez et al. (2019) to account for stain variation from multiple acquisition centers. A ResNet-50 (He et al., 2016) pre-trained on DP images via BYOL self-supervised learning strategy (Grill et al., 2020; Abbasi-Sureshjani et al., 2021) was used to extract intermediate features and the backbone weights remained frozen for computational efficiency. The attention network in attention MIL had 5 fully connected layers, followed by batch normalization and dropout (p=0.5). The transformer used Nystrom approximation (Xiong et al., 2021) for Self-Attention (SA) with 3 layers and 8 heads in each multi-head SA block. The main-task classifier had 2 fully connected layers. The discriminator had 3 layers in CDAN and 2 in DANN. ReLU is used as activation function. The Adam optimizer with a

learning rate of $10^{-3}$ was used. The adversarial contribution to the updates of the network parameters preceding the domain discriminator $\mathcal{G}$ was defined as $\lambda = \frac{2}{1+\exp(-\gamma p)} - 1$, with $p \in (0, 1]$ the relative progress of the training.

The UDA strategies were compared with three baselines: *source only* (S1), *target only* (S2), and *balanced data* (combining source and target data, with new stratification). Target labels were only used to settle the baseline and evaluating the adapted models (never for UDA). Model selection relied on macro-average validation F1-score. The performance on S3 was obtained using the model with the closest average F1-score to S2 in the CV experiments.

## 3. Results and Conclusion

The results in Table 1 show UDA improves the performance on the target, indicating higher retention of domain agnostic features. The F1-score gap for S2 is reduced by at least 10%, while the models still generalize to S3. The more severe gap for S2 than S3, beyond persisting staining differences after stain normalization, could be attributed to the slide thickness as explained by our pathologist. Moreover, the attention heatmaps showed the effectiveness of UDA to reduce bias towards light coloring that may be irrelevant to the network outcome.

No UDA method clearly outperformed the rest, possibly due to limited bandwidth for domain alignment with a frozen backbone. More complex methods with additional hyperparameters may be required. UDA led to a slight decline in source domain performance that can be addressed by continual learning methods such as Bándi et al. (2022).

## Acknowledgments

The authors thank the Roche Personalized Healthcare Digital Pathology Program for sponsoring project resourcing. The authors declare the following competing interests: S.A., N.S. and K.K. are Roche employees and J.G. was employed by Roche at the time of this work.

Table 1: Results of different UDA strategies on CAMELYON dataset, 5-fold CV [a]

| Method | | S1 | S1 $\rightarrow$ S2[b] | | S3 |
|---|---|---|---|---|---|
| MIL | Experiment | avg. F1 | avg. F1 | $\mathbf{F1}_{S1} - \mathbf{F1}_{S2}(\downarrow)$ | avg. F1 |
| Attention MIL | *Balanced data* | 85.2(1.0) | 80.4(3.2) | - | 87.8(2.2) |
| | *Source only* | 86.9(3.6) | 68.4(2.1) | 19.2(3.9) | 83.0 |
| | DANN @ $loc_i$ | 83.6(2.6) | 74.0(6.3) | 9.6(2.0) | 82.6 |
| | CDAN @ $loc_b$ | 86.2(6.2) | 80.2(3.0) | 6.0(3.2) | 85.2 |
| | DANN @ $loc_b$ | **87.6(4.3)** | **81.4(3.2)** | 6.2(1.1) | 86.0 |
| | CDAN @ $loc_i$ | 83.0(3.2) | 79.4(3.0) | 3.6(1.2) | 83.4 |
| | *Target only (S2)* | - | - | 88.4(6.1) | - |
| TransMIL | *Balanced data* | 88.5 | 85.1 | - | 92.9 |
| | *Source only* | 86.3(4.2) | 67.6(7.8) | 18.7(3.6) | 84.0 |
| | DANN @ $loc_b$ | 85.6(3.6) | 79.3(2.2) | 6.3(1.5) | 85.8 |
| | CDAN @ $loc_b$ | 86.4(2.1) | 79.0(2.5) | 7.0(0.4) | 85.0 |
| | *Target only (S2)* | - | 90.5(5.3) | - | - |

[a]Percentages with standard deviation. Best in bold, second underlined; [b]Arrow for adaptation direction.

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
