# OpenReview forum: "A Comparative Study of Unsupervised Adversarial Domain Adaptation Strategies in Multiple-instance Learning Frameworks for Digital Pathology"
_MIDL.io/2023/Short_Paper_Track — MIDL 2023 Short paper track Poster_

### Official Review · Reviewer_Rvob · 2023-04-14
**interesting evaluation study**

**Rating:** 6
**Confidence:** 4

**Review:**

Interesting evaluation study. For a short paper, it passes the threshold to be accepted. The authors investigate the effect of
changing the position of the domain discriminator in attention MIL which could potentially generate interesting discussions during the meeting.

---

### Official Review · Reviewer_MvZf · 2023-04-22

**Rating:** 6
**Confidence:** 4

**Review:**

This paper investigates the capabilities of (unsupervised) adversarial domain adaptation for the task of Multiple-Instance-Learning (MIL) based classification of cancer in histology imaging (CAMELYON database). The standard Domain Adversarial Networks (DAN) (Ganin 2016) and its Conditional extension (CDAN, Long 2017) are evaluated, using as backbone networks for MIL an attention based classifier and a Transformer based classifier (from Ilse 2018 and Shao 2021 respectively). The paper performs 2 experiments with one of the models (Attention MIL) to look into whether it is better to connect the adversarial network of DAN after the encoder but before the MIL encoder, or after the MIL encoder. The work finds that DAN/CDAN adaptation improves performance on the target domain.

Strengths:
- The problem of improving generalisation to unseen domains is still an open and interesting problem.

Weaknesses:

- Technical novelty is limited. The work only investigates the well established DAN framework. This framework has commonly been investigated in histology (see references below). The work does not mention works that investigated DAN with MIL models (attention/transformer etc), but a quick search gave me previous works that used DAN with Attention-based MIL too (similar to the Att MIL here).
- Literature review is limited and does not help position the paper within a gap in the literature. Even if there were previous works in this space, if the paper had referenced them, the authors could have argued that perhaps there is not too much work in a particular part of the field and therefore this work makes a contribution, but this is not done as no works are referenced e.g. in the DAN for MLI space. I provide below some well known works on DANs in medical imaging and histology, as well as domain adaptation in MIL for histology (a quick search on google with "adversarial adaptation multiple instance learning histology" returned me various results, and I provide Hashimoto 2020 as example), as well as works that investigated at what depth/layers it is useful to perform adaptation with DAN.
- The experiments are limited. The methods are only tested in one setting (train with source A, adapt with target B, test on C). In future work, further combinations (e.g. train B, adapt A, test C, etc) could be tried for strengthen generalisation of findings. Also, one of the main parts of the investigation (is it better to adapt at earlier or later layers?) is only evaluated for Attention-MIL but not for the Transformer-MIL (though text claims otherwise in end of paragr1 of Sec 2)

References:

* Early work on adaptation for histology: Lafarge et al, Domain-Adversarial Neural Networks to Address the Appearance Variability of Histopathology Images, DLMIA wshop at MICCAI 2017
* The above follows the below work, which is the first I am aware of that investigated DAN in medical imaging and at what layer depth DANs are effective (which is one of the main points of this work and such related previous investigations should be discussed): Kamnitsas et al, Unsupervised domain adaptation in brain lesion segmentation with adversarial networks, IPMI 2017
* Adaptation for MIL in histology: Hashimoto et al, Multi-scale Domain-adversarial Multiple-instance CNN for Cancer Subtype Classification with Unannotated Histopathological Images, CVPR 2020

Some minors:

"The adversarial contribution ... of the training.": Unclear, as \lambda has not been defined before. "Adversarial contribution" is a confusing term.

In table 1, Att MIL, Target only (S2), the 88.4 I think is in the wrong column?